# The Preventive and Curative Effects of Lactobacillus reuteri NK33 and Bifidobacterium adolescentis NK98 on Immobilization Stress-Induced Anxiety/Depression and Colitis in Mice

**DOI:** 10.3390/nu11040819

**Published:** 2019-04-11

**Authors:** Hyo-Min Jang, Kyung-Eon Lee, Dong-Hyun Kim

**Affiliations:** Neurobiota Research Center, Department of Life and Nanopharmaceutical Sciences, College of Pharmacy, Kyung Hee University, Kyungheedae-ro, Dongdaemun-gu, Seoul 02447, Korea; jhm0346@naver.com (H.-M.J.); mljun@nate.com (K.-E.L.)

**Keywords:** *Bifidobacterium adolescentis*, *Lactobacillus reuteri*, depression, anxiety, immobilization stress, colitis

## Abstract

The gut dysbiosis by stressors such as immobilization deteriorates psychiatric disorders through microbiota-gut-brain axis activation. To understand whether probiotics could simultaneously alleviate anxiety/depression and colitis, we examined their effects on immobilization stress (IS)-induced anxiety/depression and colitis in mice. The probiotics *Lactobacillus reuteri* NK33 and *Bifidobacterium adolescentis* NK98 were isolated from healthy human feces. Mice with anxiety/depression and colitis were prepared by IS treatment. NK33 and NK98 potently suppressed NF-κB activation in lipopolysaccharide (LPS)-induced BV-2 cells. Treatment with NK33 and/or NK98, which were orally gavaged in mice before or after IS treatment, significantly suppressed the occurrence and development of anxiety/depression, infiltration of Iba1^+^ and LPS^+^/CD11b^+^ cells (activated microglia) into the hippocampus, and corticosterone, IL-6, and LPS levels in the blood. Furthermore, they induced hippocampal BDNF expression while NF-κB activation was suppressed. NK33 and/or NK98 treatments suppressed IS-induced colon shortening, myeloperoxidase activity, infiltration of CD11b^+^/CD11c^+^ cells, and IL-6 expression in the colon. Their treatments also suppressed the IS-induced fecal Proteobacteria population and excessive LPS production. They also induced BDNF expression in LPS-induced SH-SY5Y cells in vitro. In conclusion, NK33 and NK98 synergistically alleviated the occurrence and development of anxiety/depression and colitis through the regulation of gut immune responses and microbiota composition.

## 1. Introduction

Anxiety disorder is the most common disorder characterized by somatic, emotional, cognitive, and behavioral components [1,2]. Patients with anxiety disorders progress to the depressive disorder that is a common illness worldwide [3]. Anxiety/depression was approximated in experimental rodents by treating stressors such as immobilization and forced swimming [4,5]. Exposure to stressors induces the secretion of adrenaline, noradrenaline, and glucocorticoids from the adrenal gland through the hypothalamic-pituitary-adrenal (HPA) axis activation, regulates the expression of proinflammatory cytokines such as tumor necrosis factor (TNF)-α and interleukin (IL)-6 in immune cells, shifts gut microbiota composition, and increases gut bacterial lipopolysaccharide (LPS) production [6,7,8]. However, the up-regulation of these proinflammatory cytokines suppresses the brain-derived neurotrophic factor (BDNF) expression in the hippocampus [8,9].

Gut microbiota consists of greater than 1000 species from relatively few phyla including Firmicutes, Bacteroidetes, and Proteobacteria [10,11]. Although the gut microbiota composition is inter-individually variable and changeable, a number of their functions for maintaining health status are associated with the core microbiota [11,12]. The gut microbiota and their byproducts stimulate enteric nervous and immune systems in the gastrointestinal tract, which propagates into other organs in the body by regulating the secretion of cytokines and adrenal hormones [13,14,15]. The occurrence of gut microbiota dysbiosis by stressors can disturb the host homeostasis, resulting in systemic disorders such as anxiety and depression [15,16,17]. Germ-free mice exhibit the severe anxiety-like behaviors compared with conventional mice [18]. However, the fecal transplantation of conventional mice into germ-free mice suppresses anxiety [19]. Thus, the brain can regulate gut function and microbiota composition through the nervous and immune systems, namely HPA axis, and the gut microbiota can regulate the psychiatric function through endocrine, neural, and immune responses, namely microbiota-gut-brain (MGB) axis [20,21]. Therefore, regulating the MGB axis is useful for the therapy of psychiatric disorders.

Probiotics, including bifidobacteria and lactobacilli, exhibit a variety of pharmacological activities: They alleviate constipation, diarrhea, imbalanced immune system, and psychiatric disorders [22,23]. *Lactobacillus plantarum*, which is isolated from zebrafish, attenuates anxiety in zebrafish [24]. *Lactobacillus reuteri*, which is isolated from mice, alleviates anxiety in mice [25]. *Bifidobacterium adolescentis* IM38, which is isolated from human gut microbiota, alleviates IS-induced anxiety in mice [26]. Nevertheless, the anti-depressive mechanism of probiotics is still unclear. 

Therefore, we isolated commensal *Lactobacillus reuteri* NK33 and *Bifidobacterium adolescentis* NK98 from healthy human feces and investigated their preventive and curative effects on immobilization stress (IS)-induced anxiety/depression and colitis in mice.

## 2. Materials and Methods

### 2.1. Materials

Dulbecco Modified Eagle Medium (DMEM) and corticosterone were purchased from Sigma (St. Louis, MO, USA). Antibodies were purchased from Cell Signaling Technology (Beverly, MA, USA). Enzyme-linked immunosorbent assay (ELISA) kits for corticosterone (E-EL-M0349) and IL-6 were purchased from Elabscience (Hebei, China) and eBioscience (San Diego, CA, USA), respectively.

### 2.2. Culture of Lactobacillus reuteri NK33 and Bifidobacterium adolescentis NK98 and Their Dosage Regimen

*Lactobacillus reuteri* NK33 and *Bifidobacterium adolescentis* NK98 were isolated from fresh human feces and identified, as previously reported [26] and deposited in the Korea Culture Center for Microorganisms (KCCM 12297 and KCTC12090). Isolated probiotics were cultured in general media for probiotics such as De Man, Rogosa and Sharpe (MRS) broth (Becton, Dickinson and Company, Radnor, PA, USA). Cultured cells were centrifuged (5000× *g*, 20 min, 20 °C), washed with saline, and suspended in saline (for in vitro experiments) or 1% maltose (for in vivo experiments).

To decide the dose of NK33 and NK98 in mouse experiments, they (1 × 10^7^, 1 × 10^8^, and 1 × 10^9^ CFU/mouse/day) were orally given for 5 days in IS-treated mice and their anti-depressive effects were evaluated in the elevated plus maze task, as previously reported [25]. Among these, treatment with NK33 or NK98 at a dose of 1 × 10^9^ CFU/mouse/day strongly alleviated IS-induced anxiety-like behaviors. Therefore, we orally gavaged NK3 or NK49 at a dose of 1 × 10^9^ CFU/mouse/day for the further in vivo study. 

### 2.3. Culture of BV2 and SH-SY5Y Cells

Human neuroblastoma SH-SY5Y and murine microglial BV-2 cells (Korea Cell Line Bank, Seoul, Korea) were cultured at 37 °C in a 95% air/5% CO_2_ atmosphere in DMEM containing 5% fetal bovine serum and 1% antibiotic-antimycotic [27]. For the assay of BDNF expression, SH-SY5Y cells (1 × 10^6^ cells/mL) were incubated with LPS (100 ng/mL, purified from *Escherichia coli* O111:B4, Sigma) in the absence or presence of test probiotics for 24 h. For the assay of IL-6 expression and NF-κB activation, BV-2 cells (1 × 10^6^ cells/mL) were incubated with LPS (100 ng/mL) in the absence or presence of test probiotics for 1.5 h (for NF-κB activation) or 20 h (for IL-6 expression). Protein expression levels were assayed by immunoblotting and ELISA.

### 2.4. Animals

C57BL/6 mice (male, 5 weeks old, 19–21 g) were supplied from Orient Bio (Seongnam-shi, Korea) and adapted for 7 days before experiments. All animals were maintained in wire cages under standard conditions of constant temperature (20 ± 2 °C), humidity (50% ± 10%) and lighting (12 h/day). All mice were fed standard laboratory chow and tap water ad libitum.

Animal experiments were conducted according to the National Institute of Health (NIH) and University Guide for Laboratory Animal Care and Usage. All animal experimental procedures were approved by the Institutional Animal Care and Use Committee of the University (IACUC No KUASP(SE)-17-146-1).

### 2.5. Preparation of Mice with Anxiety/Depression and Colitis

To examine the curative effects of probiotics on anxiety/depression, mice were randomly assigned to six groups (NC, C, PC, NK33, NK98, or Mix) of seven mice each. First, mice from the PC, NK33, NK98, Mix, and C groups were exposed to IS and test agents (C, vehicle [1% maltose]; NK33, 1 × 10^9^ CFU/mouse/day of NK33; NK98, 1 × 10^9^ CFU/mouse/day of NK98; Mix, 1 × 10^9^ CFU/mouse/day of the (1:1) mixture of NK33 and NK98]; and PC, 1 mg/kg/day of buspirone) either orally (for NK33, NK98, and Mix) or intraperitoneally (for buspirone) administered for 5 days, 24 h after the final treatment with IS. The normal control group (NC), not exposed to IS, was treated with 1% maltose in place of test agents. Behaviors and biochemical markers were assayed 20 h after the final treatment. Exposure of mice to IS was performed for 12 h once a day for 2 days using a conical tube-like instrument (2.5 cm in diameter, 7.5 cm in length) with a 0.25-cm-diameter hole on the center of the tube), as previously reported [25,28]. 

To examine the preventive effects of probiotics on anxiety/depression and colitis, mice were randomly assigned to six groups of seven mice each. First, test agents (C, vehicle [1% maltose]; NK33, 1 × 10^9^ CFU/mouse/day of NK33; NK98, 1 × 10^9^ CFU/mouse/day of NK33; Mix, 1 × 10^9^ CFU/mouse/day of the (1:1) mixture of NK33 and NK98]; and PC, 1 mg/kg/day of buspirone) were orally gavaged or intraperitoneally injected into the mice daily for 5 days. Mice from the PC, NK33, NK98, Mix, and C groups were exposed to IS for 2 h from 24 h after the final treatment with test agents, as previously reported [28]. Normal control group (NC), not exposed to IS, was orally treated with 1% maltose in place of test agents. Behaviors and biochemical markers were assayed 20 h after the final IS treatment.

### 2.6. Behavioral Tasks

The elevated plus maze task was carried out in the plus-maze apparatus, which consisted of two open (30 cm × 7 cm) and two enclosed arms (30 × 7 cm) with 20-cm-high walls extending from a central platform (7 cm × 7 cm) on a single central support to a height of 60 cm above the floor for 5 min, as previously reported [25]. The light/dark transition task was carried out in the light/dark box apparatus (45 cm × 25 cm × 25 cm), which consisted of two chambers (black and white polywoods [walls] and Plexiglass [floor] connected by an opening (7.5 cm × 7.5 cm) located at floor level in the center of the dividing wall) for 5 min, as previously reported [25]. A tail suspension test (TST) was measured according to the method of Dunn and Swiergiel [29]. Mice were suspended on the edge of a table 30 cm above the floor by taping 1 cm from the tail tip. Immobility time was measured for 5 min. When the mice did not move and passively hung, mice were judged to be immobile. A forced swimming test (FST) was performed in a round transparent plastic jar (20 cm × 40 cm) containing fresh water (25 °C) of 25 cm height according to the method of Dunn and Swiergiel [29]. Immobility time was measured for 5 min. When the mice remained floating in the water without movement, mice were judged to be immobile.

### 2.7. Immunobloting and ELISA

Hippocampus, colon tissues, SH-SY5Y, and BV2 cells were lysed with lysis RIPA buffer, which contained 1% phosphatase and protease inhibitor cocktails and centrifuged (10,000× *g*, 10 min, 4 °C) [30]. For the ELISA, the supernatants of the hippocampus and colon tissues, cultured cells, and blood, which was centrifuged (3000× *g*, 5 min, 4 °C), were transferred to a 96-well plate. Corticosterone and cytokine concentrations were assayed using their ELISA kits. For the immunoblotting analysis, the supernatant (proteins) was electrophoresed, as previously reported [30]. Electrophoresed proteins were transferred to a membrane, blocked with 5% non-fat dried-milk proteins, and probed with the corresponding antibodies. The membrane was incubated with horseradish peroxidase-conjugated secondary antibodies. Proteins were visualized with an enhanced chemiluminescence detection kit. 

### 2.8. Immunofluorescence Assay

The immunofluorescence assays of brains and colons were performed, as previously reported [30]. Briefly, mice were transcardiacally perfused with 4% paraformaldehyde. Brains and colons were removed, fixed with 4% paraformaldehyde for 4 h, cytoprotected in a 30% sucrose solution, and frozen. The frozen tissues were cut using a cryostat, incubated for 16 h at 4 °C with the Iba1 antibody for microglia, LPS and CD11b antibodies for LPS^+^/CD11b^+^ cells (activated microglia), and CD11b and CD11c antibodies for CD11b^+^/CD11c^+^ cells (dendritic cells and macrophages), and incubated with Alexa Fluor 488 (1:1000, Invitrogen, Carlsbad, CA, USA) or Alexa Fluor 594 (1:500, Abcam, Cambridge, UK) conjugated secondary antibodies. The nuclei staining was performed using 4’,6-diamidino-2-phenylindole (DAPI). The immunostained sections were observed using a confocal laser microscope. 

### 2.9. Quantitative Polymerase Chain Reaction (qPCR)

To analyze fecal microbiota, total DNA (0.1 μg) was isolated from mouse feces and qPCR was carried out with SYBER premix in a Takara thermal cycler, as previously reported [30]. The thermal cycling condition was as follows: Initial denaturation was 95 °C for 30 s; cycling, 43 s; denaturation, 95 °C for 5 s; annealing, 63 °C for 30 s; and extension, 72 °C for 30 s. The bacterial population level was calculated relative to 16S ribosomal RNA by using Microsoft Excel (Microsoft, Redmond, WA, USA). Abundance (%) was [each bacterial population in each feces]/[each bacterial population in the highest one] × 100. Primers are indicated in Table 1.

### 2.10. Limulus Amoebocyte Lysate Assay

The endotoxin concentrations of feces and blood were assayed, as previously reported [31]. Briefly, mouse fresh feces (20 mg) were suspended in sterilized saline (30 mL), sonicated for 30 min at 4°C, centrifuged (400× *g*, 15 min, 4 °C), and sequentially filtrated through 0.45 µm and 0.22 µm membrane filters. The filtrated solution was heated for 10 min at 70 °C. Mouse bloods were centrifuged (3000× *g*, 5 min, 4 °C). The plasma was diluted 10-fold, heated at 70 °C for 10 min, and centrifuged (3000× *g*, 10 min, 20 °C). Their endotoxin concentrations were assayed using a LAL assay kit (Cape Cod Inc., East Falmouth, MA, USA).

### 2.11. Statistical Analysis

All experimental values are indicated as the mean ± standard error of mean (SEM). Their significant differences were analyzed using a one-way analysis of variance (ANOVA) followed by a Duncan multiple range test (*p* < 0.05).

## 3. Results

### 3.1. The Curative Effects of NK33 and NK98 against IS-Induced Anxiety/Depression and Colitis in Mice

In order to investigate whether probiotics arising from the human gut microbiota could simultaneously alleviate anxiety/depression and colitis, we first isolated probiotics from human feces and examined their anti-inflammatory activities in LPS-treated BV-2 cells. Of tested probiotics, NK33 and NK98 strongly inhibited NF-κB activation and IL-6 expression in LPS-treated BV-2 cells (Figure 1). These probiotics were identified as *Lactobacillus reuteri* and *Bifidobacterium adolescentis*, based on the results of Gram staining, API 50 CHL kit (BNFKorea, Seoul, Korea), and 16S rDNA sequencing, respectively.

Next, we examined whether NK33 and NK98 could alleviate IS-induced anxiety/depression in mice. Exposure to IS significantly decreased the time spent in open arms (OT) and open arm entries (OE) during the elevated plus maze task to 37.3% [F(1,12) = 22.058, *p* < 0.05] and 73.0% [F(1,12) = 10.052, *p* < 0.05] of control mice, respectively (Figure 2A,B). Furthermore, IS exposure suppressed the time spent in the light compartment (TL) and number of transitions into the light dark compartment (NT) in the light/dark transition task and increased immobility in the tail suspension and forced swimming tasks. However, oral administration of NK33 or NK98 significantly mitigated OT in the elevated plus maze task to 114.8% [F(1,12) = 16.694, *p* < 0.05] and 107.0% [F(1,12) = 23.163, *p* < 0.001] of control mice, respectively, OE in the elevated plus maze task to 99.2% [F(1,12) = 7.670, *p* < 0.05] and 93.2% [F(1,12) = 5.243, *p* < 0.05] of control mice, respectively and TL in the light/dark transition task to 89.5% [F(1,12) = 8.856, *p* < 0.05] and 81.0% [F(1,12) = 4.956, *p* < 0.05] of control mice, respectively. NK33 and NK98 treatment suppressed immobility in the forced swimming task to 80.2% [F(1,12) = 6.951, *p* < 0.05] and 81.6% [F(1,12) = 6.214, *p* < 0.05] of mice treated with IS alone, respectively, and immobility in the tail suspension task to 67.3% [F(1,12) = 4.578, *p* = 0.054] and 80.59% [F(1,12) = 15.222, *p* < 0.05] of mice treated with IS alone, respectively, while NT in the light/dark transition task was not significantly influenced (Figure 2C–E). Treatment with Mix, the mixture of NK33 and NK98, also additively or synergistically alleviated anxiety/depression-like behaviors. IS exposure also inhibited hippocampal BDNF expression and CREB phosphorylation and increased hippocampal NF-κB activation (Figure 2F). Furthermore, IS exposure increased the infiltration of activated microglial cells into the hippocampus: It increased Iba1^+^ and LPS^+^/CD11b^+^ cells in the hippocampus (Figure 2G,H). However, treatment with NK33, NK98, or Mix inhibited the NF-κB activation and activated microglial cell infiltration into the hippocampus and induced hippocampal BDNF expression and CREB phosphorylation in IS-exposed mice (Figure 2F). IS exposure also increased blood corticosterone, IL-6, and LPS levels (Figure 2I–K). NK33, NK98, or Mix treatment reduced IS-induced IL-6, corticosterone, and LPS levels.

IS exposure significantly caused colitis in mice. Thus, treatment with IS caused colon shortening and induced colonic myeloperoxidase activity and NF-κB activation (Figure 3A–F). Furthermore, treatment with IS significantly increased the CD11b^+^ and/or CD11c^+^ cell (DCs and macrophages) infiltration into the colon (Figure 3G). Treatment with NK33, NK98, or Mix significantly alleviated IS-induced colon shortening and macroscopic score and suppressed myeloperoxidase activity, IL-6, IL-1β, and COX-2 expression, NF-κB activation, and CD11b^+^ and/or CD11c^+^ cell infiltration: They synergistically suppressed IL-1β expression, myeloperoxidase activity, and CD11b^+^/CD11c^+^ cell infiltration. Furthermore, IS exposure increased the population of Proteobacteria and reduced the populations of Firmicutes and Actinobacteria in the fecal microbiota. However, NK33, NK98, or Mix treatment increased IS-suppressed Bacteroidetes, Firmicutes, and Actinobacteria populations and suppressed IS-induced Proteobacteria population (Figure 3H). Furthermore, they also inhibited the fecal LPS level production in IS-induced mice (Figure 3I).

### 3.2. The Preventive Effects of NK33 and NK98 on IS-Induced Anxiety/Depression in Mice

Next, to evaluate the preventive effects of NK33, NK98, or Mix on IS-induced anxiety/depression, we orally administered probiotics daily for 5 days, then exposed IS, and examined their effects on the occurrence and development of anxiety/depression and colitis in mice (Figure 4). Exposure to IS significantly decreased OT and OE during the elevated plus maze task to 37.7% [F(1,12) = 25.676, *p* < 0.001] and 69.0% [F(1,12) = 10.090, *p* < 0.05] of control mice, respectively (Figure 4A,B). Exposure of mice to IS also significantly suppressed TL and NT in the light/dark transition task and increased immobility in tail suspension and forced swimming tasks (Figure 4C–E). However, pretreatment with NK33 or NK98 prevented the occurrence of anxiety/depression: They significantly prevented OT in the elevated plus maze task to 73.0% [F(1,12) = 10.982, *p* < 0.05] and 86.5% [F(1,12) = 16.004, *p* < 0.05] of control mice, respectively, OE in the elevated plus maze task to 105.0% [F(1,12) = 28.477, *p* < 0.001] and 97.2% [F(1,12) = 36.363, *p* < 0.001] of control mice, respectively, TL in the light/dark transition task to 95.2% [F(1,12) = 20.158, *p* < 0.05] and 92.3% [F(1,12) = 16.979, *p* < 0.05] of control mice, respectively, NT in the light/dark transition task to 82.1% [F(1,12) = 9.757, *p* < 0.05] and 78.6% [F(1,12) = 5.154, *p* < 0.05] of control mice, respectively. NK33 and NK98 reduced immobility in the forced swimming task to 71.9% [F(1,12) = 7.944, *p* < 0.05] and 59.4% [F(1,12) = 15.222, *p* < 0.05] of mice treated with IS alone, respectively, and immobility in the tail suspension task to 72.9% [F(1,12) = 4.578, *p* = 0.054] and 69.6% [F(1,12) = 4.5613, *p* = 0.054] of mice treated with IS alone, respectively. IS exposure suppressed BDNF expression and CREB phosphorylation in the hippocampus and induced NF-κB activation (Figure 4F). IS exposure also increased the infiltration of activated microglial cells into the hippocampus: It increased Iba1^+^ and LPS^+^/CD11b^+^ cells in the brain (Figure 4G,H). However, oral administration of NK33, NK98, or Mix protected the suppression of BDNF expression, the induction of NF-κB activation, and the infiltration of Iba1^+^ and LPS^+^/CD11b^+^ cells (activated microglial cells) by IS exposure. IS exposure also increased corticosterone, IL-6, and LPS levels in the blood of mice (Figure 4I–K). However, pretreatment with NK33, NK98, or Mix significantly protected the induction of corticosterone, IL-6, and LPS levels by IS exposure.

Exposure to IS alone significantly caused colitis: It induced colon shortening, colonic myeloperoxidase activity, proinflammatory cytokine IL-6 and IL-1β expression, and NF-κB activation, and COX-2 expression (Figure 5A–F). IS treatment also increased the CD11b^+^ and/or CD11c^+^ cell infiltration into the colon (Figure 5G). However, pretreatment with NK33, NK98, or Mix significantly prevented the IS-inducible colon shortening, macroscopic score, myeloperoxidase activity, IL-6 and COX-2 expression, and NF-κB activation, and CD11b^+^ and/or CD11c^+^ cell infiltration into the colon. 

Furthermore, IS exposure increased the Proteobacteria population and reduced the Firmicutes and Actinobacteria populations in the feces (Figure 5H). However, pretreatment with NK33, NK98, or Mix protected the reduction of Firmicutes and Actinobacteria populations by IS exposure and suppressed the increase of the Proteobacteria population by IS exposure. Furthermore, they protected the IS-inducible fecal LPS production (Figure 5I).

### 3.3. NK33 and NK98 Induced BDNF Expression and CREB Phosphorylation in LPS-Stimulated SH-SY5Y Cells

Next, we examined whether NK33 and NK98 could induce CREB phosphorylation and BDNF expression in LPS-stimulated SH-SY5Y cells (Figure 6). NK33 and NK98 potently increased LPS-suppressed CREB phosphorylation as well as BDNF expression. 

## 4. Discussion

The intolerable exposure to stressors such as immobilization disrupts the gut immune system and microbiota composition through the HPA axis, leading to the occurrence of gut inflammation and psychiatric disorders [4,25,32]. The exposure to stressors increases the occurrence of anxiety in germ-free or antibiotic-treated mice more sensitively than in specific pathogen-free mice with gut microbiota [18,25,33]. The fecal transplantation of SPF mice into germ-free mice reduces the occurrence of anxiety [19]. First, stressor exposure induces TNF-α and IL-6 expression in the brain, blood, and intestine with the secretion of adrenaline, noradrenaline, and glucocorticoids from the adrenal gland, leading to the increased gut membrane permeability with gut inflammation [25,34]. Second, it increases the Proteobacteria population in the gut microbiota composition and gut bacterial LPS production [25,35,36]. The disruption of gut microbiota and membrane permeability by stressors elevates the blood LPS level [30]. The excessive exposure to LPS induces TNF-α expression in the brain and suppresses BDNF expression, leading to the occurrence of anxiety/depression [37,38]. 

In the present study, we found that the exposure of mice to IS caused psychiatric disorders, anxiety and depression, hippocampal inflammation, and gut microbiota dysbiosis. Oral administration of NK33 and/or NK98 significantly suppressed the IS-induced Proteobacteria population and gut bacterial LPS production. Their treatments also alleviated colitis: They inhibited IS-induced colon shortening, myeloperoxidase activity, and macrophage and DC infiltration into the gastrointestinal tract. Furthermore, they suppressed blood LPS levels. Jang et al. also reported that IS exposure suppressed gut tight junction protein expression in the gut and brain and increased blood LPS levels by increasing gut membrane permeability [30]. These results suggest that NK33 and NK98 can reduce blood LPS levels by suppressing gut permeability through the inhibition of gut inflammation and bacterial LPS production. 

NK33 and NK98 also lowered IS-induced blood IL-6 and corticosterone levels in mice. They reduced IS-induced Iba1+ and LPS^+^/CD11b^+^ cell (activated microglia) populations in the hippocampus. Oh et al. reported that the suppression of IL-6 expression by phytochemicals induced the attenuation of anxiety in mice [39]. Corticosterone, IL-6, and TNF-α were highly expressed in patients with anxiety/depression [40,41,42]. These results suggest that NK33 and NK98 can inhibit the activation of the HPA axis, which stimulates the secretion of adrenocorticotrophin and corticosterone from the adrenal gland and proinflammatory cytokine IL-6 and TNF-α from immune cells. Furthermore, we found that NK33 and NK98 induced BDNF expression and suppressed NF-κB activation in the hippocampus. They potently suppressed IS-induced anxiety/depression-like behaviors. Lee et al. reported that the exposure to LPS suppressed NF-κB-mediated BDNF expression in the hippocampus [27]. NK33 and NK98 inhibited NF-κB in LPS-induced BV-2 cells and increased BDNF expression in LPS-stimulated SH-SY5Y cells. These results suggest that NK33 and NK98 alleviated the suppression of NF-κB-mediated BDNF expression in the hippocampus with the regulation of LPS infiltration into the brain, resulting in the attenuation of anxiety and depression. 

Moreover, the disruption of gut microbiota by IS exposure resulted in gut inflammation as well as anxiety/depression. NK33 and NK98 alleviated IS-induced anxiety and depression with the suppression of gut inflammation and microbiota disruption. Moreover, they reduced IL-6, TNF-α, and corticosterone levels, suppressed NF-κB activation in the colon and blood, and induced BDNF expression in the hippocampus. *Lactobacillus plantarum, Bifidobacterium longum* NCC3001, and *Bifidobacterium infantis* suppress anxiety-like behavior by restoring noradrenaline levels, inducing GABA production, or protecting gut dysbiosis, respectively [24,43,44]. *Bifidobacterium adolescentis* IM38, a human gut bacterium, inhibits IS-induced anxiety by regulating the GABA_A_ receptor [26]. *Lactobacillus johnsonii*, a commensal gut bacterium of mice, suppresses anxiety in IS-exposed mice by inhibiting gut bacterial LPS production [35]. *Lactobacillus rhamnosus* HN001 reduced depression and anxiety in the postpartum period of women [45]. *Lactobacillus helveticus* NS8 also improves depression-like behaviors in chronic restraint stress-exposed rats [46]. The gut microbiota is bidirectionally connected to the brain: Dysfunction of the HPA axis by stressors can cause gut microbiota dysbiosis and psychiatric disorders and probiotics can alleviate anxiety [47,48]. These results suggest that probiotics can alleviate anxiety and depression by suppressing gut dysbiosis.

## 5. Conclusions

NK33 and NK98 additively or synergistically prevented and alleviated anxiety and depression by alleviating gut dysbiosis through the suppression of the Proteobacteria population and gut microbiota LPS production.

## Figures and Tables

**Figure 1 nutrients-11-00819-f001:**
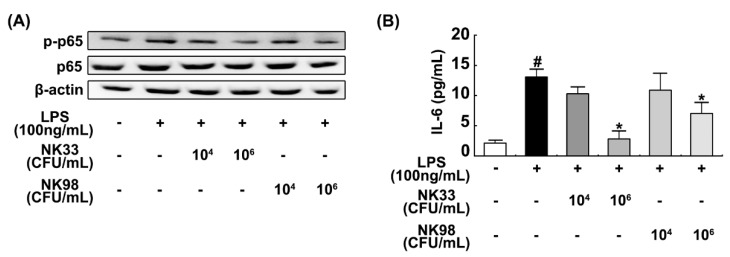
*Lactobacillus reuteri* NK33 and *Bifidobacterium adolescentis* NK98 inhibited NF-κB activation (**A**) and IL-6 expression in LPS-treated BV-2 cells (**B**). Cells (1 × 10^6^/mL) were incubated with NK33 or NK98 (1 × 10^4^ or 1 × 10^6^ CFU/mL) in the presence of LPS for 90 min (for NF-κB) or 20 h (for IL-6). IL-6 was measured with an ELISA kit. p-p65 and p65 (NF-κB) were measured by immunoblotting. Data values are indicated as mean ± standard error of mean (SEM) (*n* = 4). ^#^
*p* < 0.05 vs. group not treated with LPS; * *p* < 0.05 vs. LPS alone treated group. LPS, lipopolysaccharide.

**Figure 2 nutrients-11-00819-f002:**
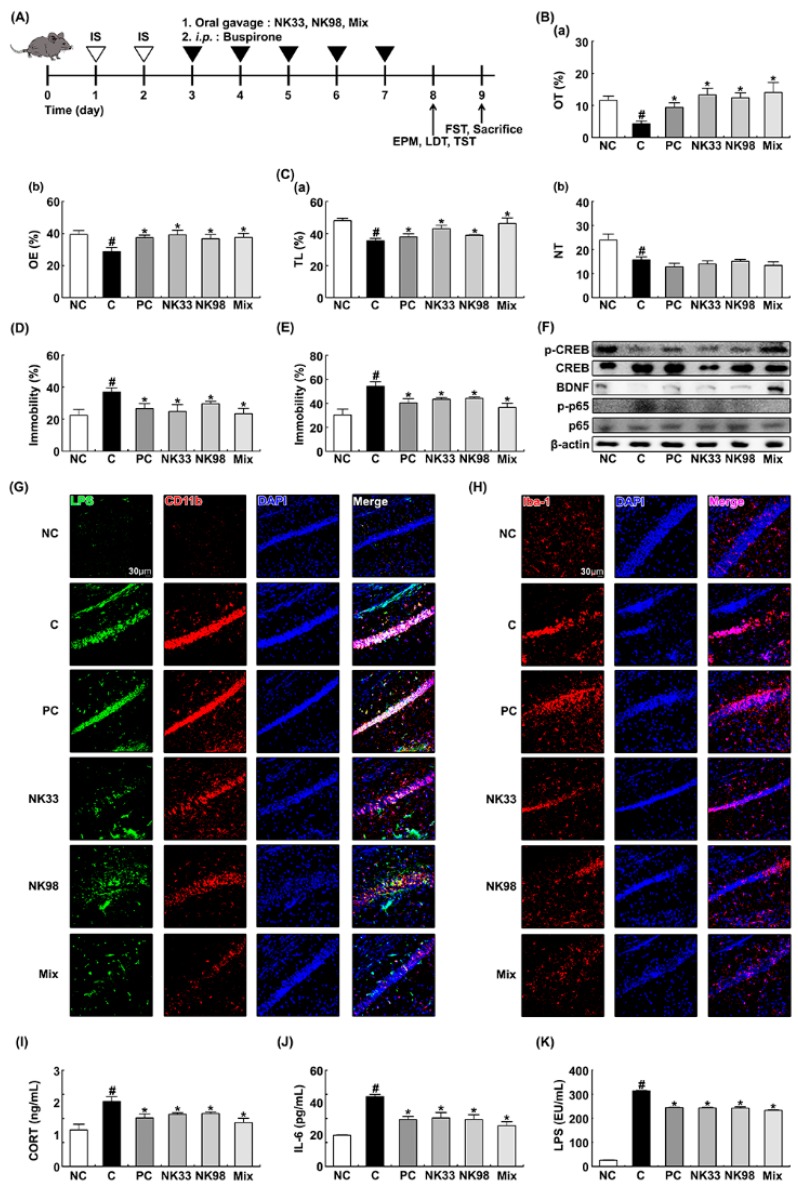
Oral administration of *Lactobacillus reuteri* NK33 and/or *Bifidobacterium adolescentis* NK98 attenuated IS-induced anxiety/depression and hippocampal inflammation in mice. (**A**) Procedure. Effects on anxiety/depression-like behaviors in elevated plus maze (**B**: (**a**), OT; (**b**), OE), LDT (**C**: (**a**), TL; (**b**), NT), tail suspension (**D**), and forced swimming (**E**) tasks. (**F**) Effects on hippocampal BDNF expression, and CREB and NF-κB activation. Effects on the infiltration of LPS^+^/CD11b^+^ (**G**) and Iba1^+^ cells (**H**) into the hippocampus. Effects on blood corticosterone (CORT, **I**), IL-6 (**J**), and LPS levels (**K**). Mice were exposed to IS and test agents (C, vehicle [1% maltose]; NK33, 1 × 10^9^ CFU/mouse/day of NK33; NK98, 1 × 10^9^ CFU/mouse/day of NK33; Mix, 1 × 10^9^ CFU/mouse/day of the (1:1) mixture of NK33 and NK98]; and PC, 1 mg/kg/day of buspirone) were gavaged (for vehicle, NK33, and NK98) or intraperitoneally injected (for buspirone) daily for 5 days. Normal control group (NC), not exposed to IS, was treated with 1% maltose instead of test agents. Hippocampal p65, p-p65, CREB, p-CREB, BDNF, and β-actin were analyzed by immunoblotting. Blood IL-6, corticosterone, and LPS were assayed by ELISA kits. Iba1^+^ and LPS^+^/CD11b^+^ cells were measured using a confocal microscope. Data values were indicated as mean ± SEM (*n* = 7). ^#^
*p* < 0.05 vs. NC group. * *p* < 0.05 vs. IS alone treated group.

**Figure 3 nutrients-11-00819-f003:**
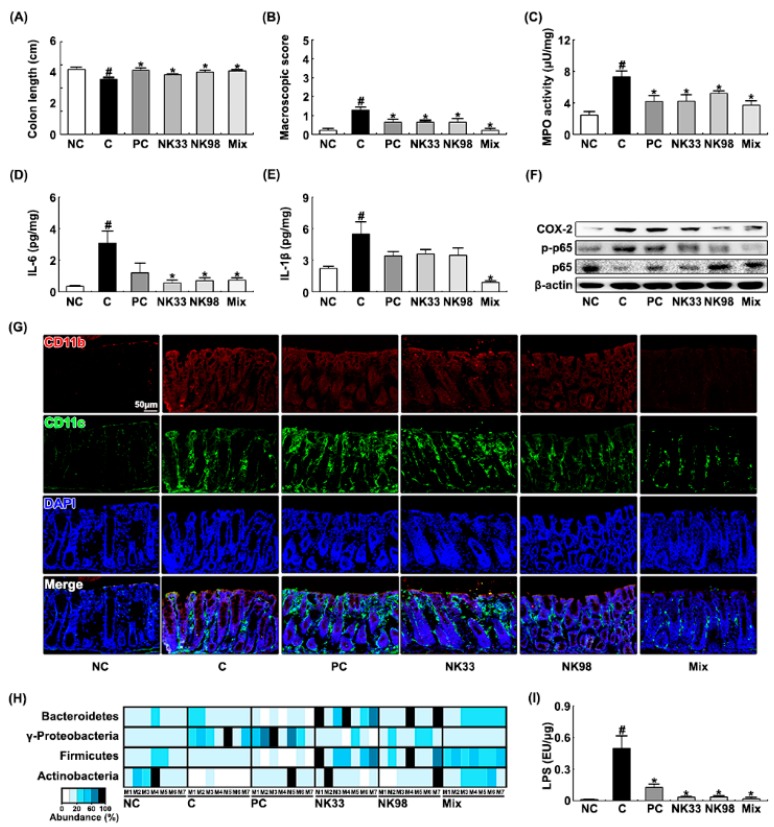
Oral administration of *Lactobacillus reuteri* NK33 and/or *Bifidobacterium adolescentis* NK98 alleviated immobilization stress (IS)-induced colitis in mice. Effects on the colon length (**A**), macroscopic score (**B**), myeloperoxidase (MPO) activity (**C**), IL-6 (**D**), IL-1β (**E**), and COX-2 expression, and NF-κB activation (**F**). Effects on the infiltration of CD11b^+^/CD11c^+^ cells into the colon (**G**), fecal microbiota composition (**H**), and fecal LPS levels (**I**). First, mice were exposed to IS and test agents (C, vehicle [1% maltose]; NK33, 1 × 10^9^ CFU/mouse/day of NK33; NK98, 1 × 10^9^ CFU/mouse/day of NK33; Mix, 1 × 10^9^ CFU/mouse/day of the (1:1) mixture of NK33 and NK98]; and PC, 1 mg/kg/day of buspirone) were gavaged (for vehicle, NK33, and NK98) or intraperitoneally injected (for buspirone) daily for 5 days. Normal control group (NC), not exposed to IS, was treated with 1% maltose instead of test agents. Colonic p65, p-p65, COX-2, and β-actin were analyzed by immunoblotting. CD11b^+^ and CD11c^+^ cell populations were assayed using a confocal microscope. Fecal bacteria were assayed by qPCR. Fecal LPS assayed by ELISA kit. Data values are indicated as mean ± SEM (*n* = 7). ^#^
*p* < 0.05 vs. NC group. * *p* < 0.05 vs. IS group.

**Figure 4 nutrients-11-00819-f004:**
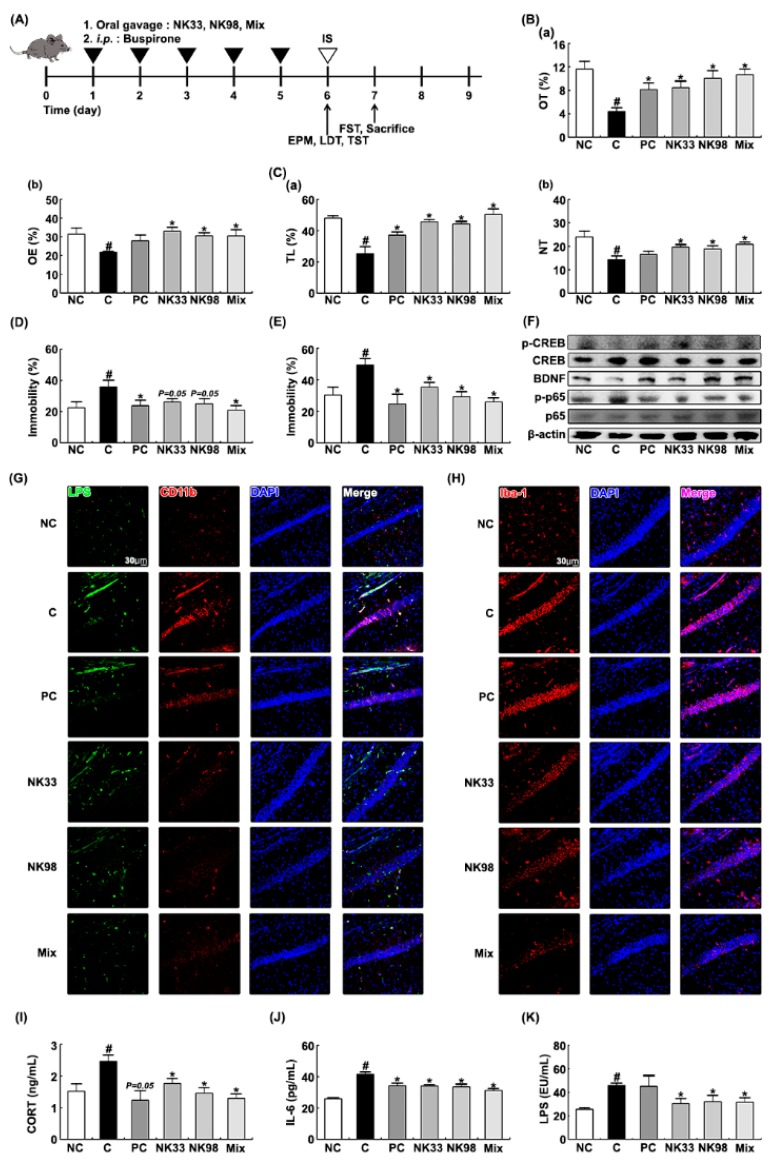
Pretreatment with *Lactobacillus reuteri* NK33 and/or *Bifidobacterium adolescentis* NK98 protected IS-induced anxiety and hippocampal inflammation in mice. (**A**) Procedure. Effects on anxiety-like behaviors in elevated plus maze (**B**: (**a**), OT; (**b**), OE), LDT (**C**: (**a**), TL; (**b**), NT), tail suspension (**D**), and forced swimming (**E**) tasks. (**F**) Effects on hippocampal BDNF expression and CREB and NF-κB activation. Effects on the infiltration of LPS^+^/CD11b^+^ (**G**) and Iba1^+^ cells (**H**) into the hippocampus. Effects on blood corticoterone (CORT, **I**), IL-6 (**J**), and LPS levels (**K**). Test agents (C, vehicle [1% maltose]; NK33, 1 × 10^9^ CFU/mouse/day of NK33; NK98, 1 × 10^9^ CFU/mouse/day of NK33; Mix, 1 × 10^9^ CFU/mouse/day of the (1:1) mixture of NK33 and NK98]; and PC, 1 mg/kg/day of buspirone) were gavaged (for vehicle, NK33, and NK98) or intraperitoneally injected (for buspirone) daily for 5 days and IS then exposed to mice. Normal control group (NC), not exposed to IS, were treated with 1% maltose instead of test agents. Hippocampal p65, p-p65, CREB, p-CREB, BDNF, and β-actin were assayed by immunoblotting. Blood corticosterone, IL-6, and LPS were assayed by ELISA kits. Iba1^+^ and LPS^+^/CD11b^+^ cells were measured using a confocal microscope. Data values are indicated as mean ± SEM (*n* = 7). ^#^
*p* < 0.05 vs. NC group. * *p* < 0.05 vs. IS group.

**Figure 5 nutrients-11-00819-f005:**
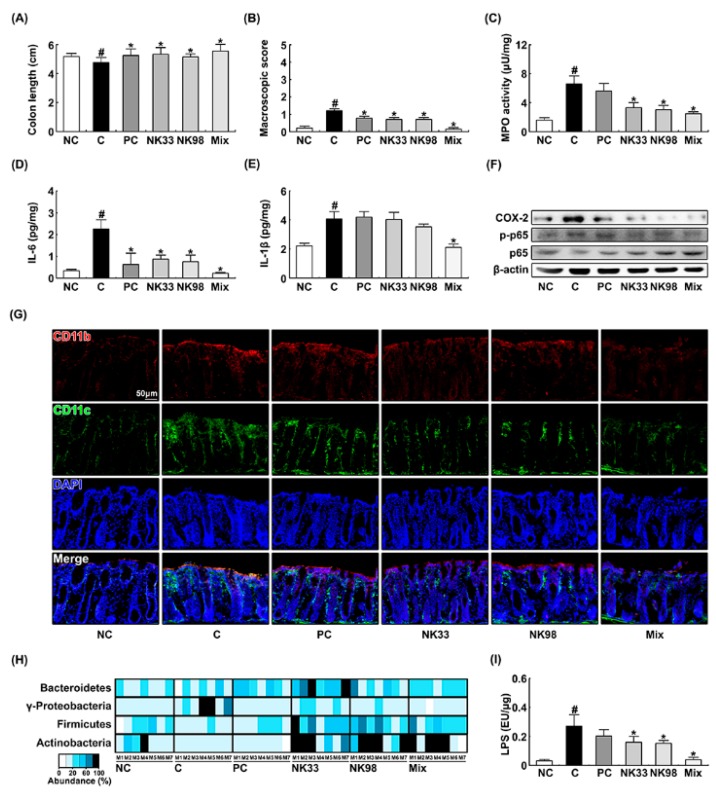
Pretreatment with *Lactobacillus reuteri* NK33 and/or *Bifidobacterium adolescentis* NK98 protected IS-induced colitis in mice. Effects on colon length (**A**), macroscopic score (**B**), colonic myeloperoxidase (MPO) activity (**C**), IL-6 (**D**), IL-1β (**E**), and COX-2 expression and NF-κB activation (**F**). (**G**) Effects on the CD11b^+^/CD11c^+^ cell infiltration into the colon. Effects on fecal microbiota composition (**H**) and fecal LPS levels (**I**). Test agents (C, vehicle [1% maltose]; NK33, 1 × 10^9^ CFU/mouse/day of NK33; NK98, 1 × 10^9^ CFU/mouse/day of NK33; Mix, 1 × 10^9^ CFU/mouse/day of the (1:1) mixture of NK33 and NK98]; and PC, 1 mg/kg/day of buspirone) were gavaged (for vehicle, NK33, and NK98) or intraperitoneally injected (for buspirone) daily for 5 days and IS then exposed to mice. Normal control group (NC), not exposed to IS, were treated with 1% maltose instead of test agents. Colonic p65, p-p65, COX-2, and β-actin were analyzed by immunoblotting. CD11b^+^ and CD11c^+^ cells were assayed using a confocal microscope. Fecal bacteria were assayed by qPCR. Fecal LPS were assayed by ELISA kit. Data values are indicated as mean ± SEM (*n* = 7). ^#^
*p* < 0.05 vs. NC group. * *p* < 0.05 vs. IS group.

**Figure 6 nutrients-11-00819-f006:**
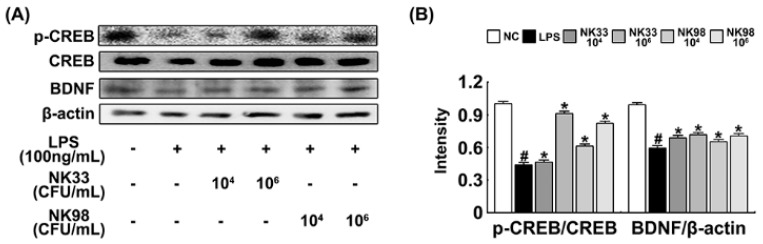
Effect of *Lactobacillus reuteri* NK33 and *Bifidobacterium adolescentis* NK98 on BDNF expression (**A**) and CREB activation in LPS-stimulated SH-SY5Y cells (**B**). Cells (1 × 10^6^) were treated with LPS in the absence or presence of NK33 or NK98 (1 × 10^4^ or 1 × 10^6^ CFU/mL) for 20 h. NC was treated with vehicle alone. BDNF, CREB, p-CREB, and β-actin were measured by immunoblotting. Data values are indicated as mean ± SEM (*n* = 4). ^#^
*p* < 0.05 vs. group not treated with LPS; * *p* < 0.05 vs. LPS alone treated group.

**Table 1 nutrients-11-00819-t001:** Primers used in the present experiments.

Phylum	Primer Sequence	
	Forward	Reverse
Firmicutes	5′-GGAGYATGTGGTTTAATTCGAAGCA-3′	5′-AGCTGACGACAACCATGCAC-3′
Bacteroidetes	5′-GTTTAATTCGATGATACGCGAG-3′	5′-TTAASCCGACACCTCACGG-3′
Actinobacteria	5′-TGTAGCGGTGGAATGCGC-3′	5′-AATTAAGCCACATGCTCCGCT-3′
δ/γ-Proteobacteria	5′-GCTAACGCATTAAGTRYCCCG-3′	5′-GCCATGCRGCACCTGTCT-3′
16S rRNA	5′-TCGTCGGCAGCGTCAGATGTGTATAA GAGACAGGTGCCAGCMGCCGCGGTAA-3′	5′-GTCTCGTGGGCTCGGAGATGTGTATA AGAGACAGGGACTACHVGGGTWTCTAAT-3′

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
