# Peer review of "The Preventive and Curative Effects of Lactobacillus reuteri NK33 and Bifidobacterium adolescentis NK98 on Immobilization Stress-Induced Anxiety/Depression and Colitis in Mice"

_nutrients, 2019, doi:10.3390/nu11040819_

Reviewer 1 Report

The authors report the preventive e curative effects of two selected strains of probiotics on a stress animal model.

The study is sufficiently interesting.

The first 3 lines of discussion are unusual and inappropriate, and should be omitted.

The discussion should be  further develop and enriched in the references.

Some grammatical errors are present.

Author Response

The authors report the preventive e curative effects of two selected strains of probiotics on a stress animal model. The study is sufficiently interesting.

→Thank you.

The first 3 lines of discussion are unusual and inappropriate, and should be omitted.

→Thank you for your comment. It was deleted.

The discussion should be further develop and enriched in the references.

→Thank you. We revised our manuscript.

Some grammatical errors are present.

→Thank you. We checked our manuscript by a native speaker (High Sierra Seoul Company, www.supreme-trans.co.kr).

Reviewer 2 Report

Overall this is a well conducted and presented project. The mouse model system requires some justification though. This especially applies to colitis. The model used is not typical of colitis. Indeed the entire aspect of colitis requires further explanation. The word colitis does not appear at all in the title of the paper. Similarly, there is no discussion of colitis in the Introduction...even though it is stated as a main aim. Curiously the word colitis does not even appear in the methods. Indeed there is no section on the induction of colitis.

The abstract needs re-writing with a Background/aims/methods/results/conclusions format. No p-values are presented.

The data would be better presented as SEM and not SD. Results are somewhat overstated (reflected in the tile) with claims of 'cure' and 'prevention'.

English grammar clearly requires attention and the first few sentences of the discussion give evidence of hasty preparation.

Author Response

Overall this is a well conducted and presented project. The mouse model system requires some justification though. This especially applies to colitis. The model used is not typical of colitis. Indeed the entire aspect of colitis requires further explanation. The word colitis does not appear at all in the title of the paper. Similarly, there is no discussion of colitis in the Introduction...even though it is stated as a main aim. Curiously the word colitis does not even appear in the methods. Indeed there is no section on the induction of colitis.

→Thank you.

The abstract needs re-writing with a Background/aims/methods/results/conclusions format. No p-values are presented.

→Thank you. We revised the abstract section of our manuscript. And p-values are indicated in the result section.

The data would be better presented as SEM and not SD. Results are somewhat overstated (reflected in the tile) with claims of 'cure' and 'prevention'.

→Thank you. We revised SD in the figures of our manuscript to SEM according to your comment.

English grammar clearly requires attention and the first few sentences of the discussion give evidence of hasty preparation.

→Thank you. We checked our manuscript by a native speaker (High Sierra Seoul Company, www.supreme-trans.co.kr).

Reviewer 3 Report

Jang et al. studies the effects of two potential probiotic strains on immobilized stress-induced anxiety and depression in mice. Oral administration of either microbes, as well as their mixture, results in both curative and preventive effects on depression/anxiety symptoms. The effect of microbes is also monitored in vitro, on BV-2 microglial cells (mouse) (IL-6 and nfkb) and  BDNF expression in SH-SY5Y cells (human). The study attempts to shed some light on cellular and molecular mechanism that may account for these observations, coroborating the link between the gut microbiota and the gut-brain  axis. Overall the study seems solid and well described.

Minors:

The manuscript contains some typos (e.g. line 45 check Firmicutes;L65 anti-inflammatory; L88 replace “like the” with “as”;  L122, in the second half of the line  “NK98” instead of NK33; L299 corticosterone, ect. ) that should be corrected.

L 65 how can you say these strains are anti-inflammatory? I guess this is based on results shown in figure 1.

L81-82 informations on microbial species identification and characterization (molecular/phenotypical) should be provided in this section (not only below in L197-199). Jang et al. 2018 describes the effect of a Bifidobactrium adolescentis, but I suppose it is a different  strain, not the same as this study…

Were the microbes used and/or their significant rDNA sequences deposited in a public repository/gene bank? (16s rDNA sequences are mentioned at L198)

L93 which kind of cells are they? Please provide more info on the type of cells and indicate also their origin (human, murine). If SH-SY5Y cells are human origin you should specify if you used same or different primary antibodies for detection of BDNF expression and CREB phosphorylation in mice hippocampus (fig 2-4) and cell cultures (fig 6).

L92-99 did you use live bacterial cells for this experiment? upon long incubation (20 h) did you monitor viability of both types of cells (microbial and animal) and or acidification of the media? Did you test potential cytoxicity of bacterial cells?

L114 the unexpert reader might like to know why you treat with buspirone.

L109-119 the one described seems the schedule depicted in fig 4A , though this figure relates to PREVENTIVE effect of treatments...by contrast the diagram of fig 2A seems to describe the plan to assess the PREVENTIVE effect of treatments (which is explained in L120-129). Is it possible that you exchanged the schedule diagram between fig 2A and 4A? Please check.

L115 126 and legend to figures: you should explain more clearly that NC group is not subjected to IS stress (but only receives vehicle).

L227 the “IS-treated control (IS) group.” Is that group indicated as “C” in the figures (also in following figures).

L281 and L310 eliminate “Experimental  schedule  (A).” as this is not present in these figures.

Fig 2A and Fig 4A: please check the schedule as they seem exchanged (see also above).

L323-24 from fig 6 one cannot see a POTENTLY increased BDNF expression... maybe oly for NK98 at higher CFU concentration this is a little more evident. Did you perform densitometric quantification of the bands?

L331-333 should be deleted

L337 abbreviations such as SPF (?) should be specified at first quoting in the text; the meaning of abbreviation is missing also for other acronyms in the text eg. See below GI (gastro-intestinal). Please check throughout the text for other possible unexplained abbreviations.

L352 GI tract

L367 or corticosterone-suppressed or rather LPS-suppresses (in fig 6 you show LPS)

Author Response

First of all, we greatly appreciate your and reviewers’ excellent suggestions. We revised our manuscript according to the your suggestions. The sentences in the manuscript revised by your comments are underlined. And the repetition rate was also reduced. The sentences revised for grammatical errors and repetition were not indicated. I hope you will consider this paper as suitable for publication.

Minors:

The manuscript contains some typos (e.g. line 45 check Firmicutes;L65 anti-inflammatory; L88 replace “like the” with “as”;  L122, in the second half of the line  “NK98” instead of NK33; L299 corticosterone, ect. ) that should be corrected.

 →Thank you. We revised it according to your comments.

L 65 how can you say these strains are anti-inflammatory? I guess this is based on results shown in figure 1.

→Thank you. We revised it according to your comments.

L81-82 informations on microbial species identification and characterization (molecular/phenotypical) should be provided in this section (not only below in L197-199). Jang et al. 2018 describes the effect of a Bifidobactrium adolescentis, but I suppose it is a different  strain, not the same as this study…

→Thank you. We revised it according to your comments.

Were the microbes used and/or their significant rDNA sequences deposited in a public repository/gene bank? (16s rDNA sequences are mentioned at L198)

→Thank you. We deposited the bacteria in KCCM.

L93 which kind of cells are they? Please provide more info on the type of cells and indicate also their origin (human, murine). If SH-SY5Y cells are human origin you should specify if you used same or different primary antibodies for detection of BDNF expression and CREB phosphorylation in mice hippocampus (fig 2-4) and cell cultures (fig 6).

→Thank you. We revised it according to your comments. We used two kinds of antibodies for BDNF (human and murine) in the previous study. Antibody for murine was working in SH-SY5Y cells. Therefore, we used the antibody for murine in the present study.

 L92-99 did you use live bacterial cells for this experiment? upon long incubation (20 h) did you monitor viability of both types of cells (microbial and animal) and or acidification of the media? Did you test potential cytoxicity of bacterial cells?

→Thank you. It was not cytotoxic under the present experimental condition.

L114 the unexpert reader might like to know why you treat with buspirone.

→Thank you. It is a well-known anxiolytic drugs. Therefore, we used it as a positive agent.

 L109-119 the one described seems the schedule depicted in fig 4A , though this figure relates to PREVENTIVE effect of treatments...by contrast the diagram of fig 2A seems to describe the plan to assess the PREVENTIVE effect of treatments (which is explained in L120-129). Is it possible that you exchanged the schedule diagram between fig 2A and 4A? Please check.

Thank you for your comment. We revised them.

L115 126 and legend to figures: you should explain more clearly that NC group is not subjected to IS stress (but only receives vehicle).

Thank you. We revised it according to your comment.

L227 the “IS-treated control (IS) group.” Is that group indicated as “C” in the figures (also in following figures).

Thank you for your comment.

L281 and L310 eliminate “Experimental  schedule  (A).” as this is not present in these figures.

Thank you for your comment. We revised it.

Fig 2A and Fig 4A: please check the schedule as they seem exchanged (see also above).

Thank you for your comment. We revised it.

 L323-24 from fig 6 one cannot see a POTENTLY increased BDNF expression... maybe oly for NK98 at higher CFU concentration this is a little more evident. Did you perform densitometric quantification of the bands?

Thank you. We added it according to your comment.

L331-333 should be deleted

Thank you for your comment. We revised it.

 L337 abbreviations such as SPF (?) should be specified at first quoting in the text; the meaning of abbreviation is missing also for other acronyms in the text eg. See below GI (gastro-intestinal). Please check throughout the text for other possible unexplained abbreviations.

Thank you for your comment. We revised it.

 L352 GI tract

Thank you for your comment. We revised it.

 L367 or corticosterone-suppressed or rather LPS-suppresses (in fig 6 you show LPS)

Thank you for your comment. We revised it.

Round  2

Reviewer 2 Report

The revised version is greatly improved. However, the revised sections still suffer from poor English grammar. For example: 'did not exposed to IS and...'.

Author Response

Thank you for your detail comments.

We revised our manuscript (L19, L45, L112, L123, L157, L217, L280, L300, L313, and L338) according to your comments. The sentences in the manuscript revised by your comments are underlined. I hope you will consider this paper as suitable for publication.